



# Estimating vertical wind power density by using tower observation and empirical models over varied desert steppe terrain in northern China

Shaohui Zhou[1], Yuanjian Yang[1*], Zhiqiu Gao[1,2*], Xingya Xi[3], and Zexia Duan[1], Yubin Li[1]

[1]Collaborative Innovation Centre on Forecast and Evaluation of Meteorological Disasters, Key Laboratory for Aerosol-Cloud-Precipitation of China Meteorological Administration, School of Atmospheric Physics, Nanjing University of Information Science and Technology, Nanjing 210044, China
[2]State Key Laboratory of Atmospheric Boundary Layer Physics and Atmospheric Chemistry, Institute of Atmospheric Physics, Chinese Academy of Sciences, Beijing 100029, China
[3]Collaborative Innovation Centre of the Atmospheric Environment and Equipment Technology, School of Environmental Science and Engineering, Nanjing University of Information Science and Technology, Nanjing 210044, China

*Correspondence to*: Yuanjian Yang (yyj1985@nuist.edu.cn) and Zhiqiu Gao (zgao@nuist.edu.cn)

**Abstract.** A complex and varied terrain has a great impact on the distribution of wind energy resources, resulting in uncertainty in accurately assessing wind energy resources. In this study, three wind speed distributions of kernel, Weibull, and Rayleigh type for estimating average wind power density were first compared by using meteorological tower data from 2018 to 2020 under varied desert steppe terrain contexts in northern China. Then three key parameters of scale factor ($c$) and shape factor ($k$) from the Weibull model and surface roughness ($z_0$) were investigated for estimating wind energy resource. The results show that the Weibull distribution is the most suitable wind speed distribution over that terrain. The scale factor ($c$) in the Weibull distribution model increases with an increase in height, exhibiting an obvious form of power function. While there were two different forms for the relationship between the shape factor ($k$) and height: i.e., the reciprocal of the quadratic function and the logarithmic function, respectively. The estimated roughness length ($z_0$) varied with the withering period, the growing period, and the lush period, which can be represented by the estimated median value in each period. The maximum and minimum values of surface roughness length over the whole period are 0.15 m and 0.12 m, respectively. The power-law model and the logarithmic model are used to estimate the average power density values at six specific heights, which show greater differences in autumn and winter, and smaller differences in spring and summer. The gradient of the increase in average power density values with height is largest in autumn and winter, and smallest in spring and summer. Our findings suggest that dynamic changes in three key parameters ($c$, $k$, and $z_0$) should be accurately considered for estimating wind energy resources under varied desert steppe terrain contexts.

## 1 Introduction

Wind energy is a renewable, environmentally friendly, and popular alternative source of clean energy (Islam et al., 2013; Gabbasa et al., 2013) and as a source of power it has a great potential (Chaurasiya et al., 2019). 93 GW of new installations



brought global cumulative wind power capacity up to 743 GW in 2020. In the onshore market, 86.9 GW was installed, an increase of 59% compared to 2019. China and the United States remain the world's largest markets for new onshore installations (2021). To use this kind of nonpolluting energy, a lot of research has been conducted through a variety of different

methods to develop an accurate and reliable wind energy evaluation model.

The wind speed probability density function can effectively characterize wind speed. Therefore, the wind speed probability density function is of great significance in wind turbine site selection, wind farm design, generator design, determination of the dominant wind direction, and evaluation of wind conversion system management and operation (Masseran, 2015; Li and Shi, 2010). Wind shows great differences with various topographies, landforms, and meteorological conditions. The magnitude

and direction of the wind speed exhibit significant differences when wind flows over rough ground or obstacles in a complex terrain. In addition, the surface topography and roughness of the area around the location of the wind measurement tower will affect the predicted wind resources (Kim and Lim, 2017). Therefore, the wind speed probability density function and roughness are important input factors in the estimation of wind energy power density.

Different distribution functions have different fitting effects on the actual wind speed values in different study areas. According

to previous studies (Lo Brano et al., 2011; Celik, 2004; Masseran et al., 2012), seven wind probability density functions have been widely used to fit the actual wind speed values: i.e., Weibull, Rayleigh, Lognormal, Gamma, Inverse Gaussian, Pearson type V, and Burr. These models exhibited different advantages and disadvantages for estimating wind probability density. For instance, (Celik, 2004) used the Weibull and Rayleigh models to perform a statistical analysis of wind energy density in southern Turkey and found that the Weibull model not only fits the measured monthly probability density distribution better

but also provides better power density estimation compared to the Rayleigh model. (Masseran et al., 2012) used nine different wind speed probability density function models to describe wind speed conditions in different regions of Malaysia and found that Gamma, Weibull, and Inverse gamma models can fit the wind speed data better. (Chang, 2011a) used six different probability density functions: namely Weibull, mixture Gamma and Weibull, mixture normal, mixture normal and Weibull, mixture Weibull, and maximum entropy principle distribution. They were tested on the wind data of three wind farms in

Taiwan and it was found that, when the current wind speed distribution is unimodal, the fitting effects of these six probability density functions are not significantly different. When the wind speed distribution is bimodal, the other five probability density functions are better than Weibull at describing wind characteristics. In addition, many other probability density functions have been invented to provide more accurate results for the estimation of wind power density in a specific area (Masseran, 2015; Carta et al., 2009; Jaramillo and Borja, 2004).

Among the above-mentioned various types of wind speed probability density functions, the Weibull and Rayleigh distributions are still the more traditional and widely applicable typical wind speed distribution forms. The key issue in the study of the Weibull distribution is how to accurately determine the values of Weibull scale factor $c$ and shape factor $k$ (Azad et al., 2014; kaplan, 2017). Generally, six different methods, i.e., graphical method (Basu et al., 2009), empirical method (Costa Rocha et al., 2012; Kaoga et al., 2014), maximum likelihood method (Andrade et al., 2014; Azad et al., 2014), energy trend method

(Chang, 2011b; Akdağ and Dinler, 2009), energy pattern method (Andrade et al., 2014), and the moment method (Azad et al.,



2014; kaplan, 2017; Costa Rocha et al., 2012), have been employed to calculate the $c$ and $k$ of the Weibull distribution model. But these methods perform differently in different regions. For instance, (kaplan, 2017) found that the energy pattern method and the moment method were the best methods between 2009 and 2013 in the Hatay and Osmaniye regions. When the time series of wind data is provided, the maximum likelihood method is more robust and accurate than other methods (Seguro and

Lambert, 2000; George, 2014). In addition, there is a strong time dependence and a high change dependence for the changes in shape factor $k$ and scale factor $c$ (Lun and Lam, 2000; Justus and Mikhail, 1976): e.g., the scale factor $c$ has a power-law functional relationship with height and the shape factor $k$ has a reciprocal logarithmic functional relationship with height. Therefore, we can explore its general laws by studying the seasonal changes and height changes in shape and scale parameters in a specific area.

Roughness length plays a key role in estimating wind energy resources. For example, (Laporte, 2010) pointed out that the roughness estimation error can cause 5% to 10% of the wind energy resource estimation error. Current wind energy resource assessment is based on measured wind data at a height of 60 to 80 m from the ground, but the actual height of the hub may be greater than these heights. Therefore, we need to combine the surface roughness length and the known wind speed value of the measured height to extrapolate the wind speed value at the height of the hub (Nayyar and Ali, 2020). Theoretically, the

surface roughness length $z_0$ is the height at which the average wind decreases to zero with height. $z_0$ varies with the underlying surface (Davenport et al., 2000; Duan et al., 2021). Currently, three approaches (the analysis method, the Charnock method, and the statistical method) have been widely applied to estimate the surface roughness length of offshore wind energy (Golbazi and Archer, 2019). Among them, the statistical method is convenient, as it needs only three layers of wind speed data. After comparing the average value and median value of roughness $z_0$, it is found that the median value is an order of magnitude

closer to the roughness length calculated from the other two methods. Therefore, when using the field measurement method to statistically determine the surface roughness length, attention should be paid to using the median value instead of the average value; otherwise huge errors will be generated when the wind speed is extrapolated to the height of the hub, which will have a huge impact on the evaluation of wind energy resources.

As an important production base of wind power energy in northern China, Inner Mongolia is under the influence of the westerly

wind all year round. The types of underlying surfaces of wind power towers in China are complex and diverse, including offshore, mountainous, urban outskirts, and grasslands. In Inner Mongolia, especially the desert grassland, the terrain is open, the vegetation is low and sparse, and its wind resources are very rich. So taking the Ningyuanbailiutu site as an example, in-depth data mining was carried out on the 4 heights of 10 m, 30 m, 50 m, and 70 m for the meteorological element data of a 100-meter wind tower from the autumn of 2018 to the summer of 2020 in Damaoqi, Baotou City, Inner Mongolia, China. The

following three steps are used to study the three important key parameters that affect the evaluation of wind energy resources: the surface roughness length $z_0$, the scale factor $c$, and the shape factor $k$ in the Weibull distribution function. Firstly, we need to determine the uniqueness and importance of the Weibull distribution function in the wind speed time series data in the Damaoqi area. This is reflected in the advances and shortcomings of the kernel distribution model, the Rayleigh distribution model, the Weibull distribution function, and the frequency distribution model using actual wind speed, which are used to





calculate the monthly, seasonal, and all-time average power densities. Secondly, by studying the monthly and seasonal changes in the surface roughness length and the changes in different incoming flow directions, we will gain a comprehensive understanding of the roughness of the site area in Inner Mongolia. Finally, by using two different models, namely the power-law model with scale parameter *c* and the logarithmic model with roughness information, the average wind power densities at six specific heights (75 m, 80 m, 85 m, 90 m, 95 m, and 100 m) per month, per season, and throughout the period are calculated. In this way, we discuss the application significance of the two models for wind energy development, and provide a scientific reference for a further understanding of the wind energy resources in the region.

## 2 Study site, data, and methods

### 2.1 Study site and data

In this study, long-term in-situ measurement was conducted in Damaoqi, Baotou City, Inner Mongolia (42 °04'25.738"N, 110 °29'2.778"E; 1376 m above sea level) from September 1, 2018 to August 31, 2020 (Figure 1). The observation wind tower is located at the northern foot of Daqing Mountain in the central area of the Inner Mongolia Plateau. Wind speed and wind direction (010C cup anemometers and 020C wind vanes, Metone, USA), atmospheric pressure (CS106 Campbell, USA), air temperature, and humidity (HC2-S3, Rotronic, Switzerland) were measured at 4 levels (i.e., 10 m, 30 m, 50 m, and 70 m) of the tower. It is surrounded by typical desert grassland. The site is characterized by a middle temperate zone and semi-arid continental climate. During the experimental period, the daily air temperature ranged between −27.3 °C and 33.9 °C, with an average value of 6.3 °C (Figure 2a). Surface-level air pressure has an inverse relation with air temperature, with an average value of 862.9 hPa (Figure 2b). In addition, the daily average relative humidity maintains a level of 41.02% and fluctuates back and forth. The average wind speed at the 70-m height is 7.61 m/s. The average daily wind speed in spring and autumn occasionally exceeds the level of 10 m/s, indicating that the site has sufficient wind resources in these two seasons (Figure 2c). The predominant wind direction was southwesterly and northwesterly during the whole observation period (Figure 2d and Figure 3).

### 2.2 Methods

### 2.2.1 Kernel, Weibull, and Rayleigh distributions

The kernel density estimator is the estimated probability density function (PDF) of a random variable. For any real values of *v*, the formula for the kernel density estimator is given by:

$$f_h(v) = \frac{1}{nh} \sum_{i=1}^{n} K\left(\frac{v-v_i}{h}\right),$$   (2.1)

where $v_1$, $v_2$, …, $v_i$ are random wind samples from an unknown distribution, *n* is the sample size, $K(\cdot)$ is the kernel smoothing function, and *h* is the bandwidth.





The probability density function of the Weibull distribution is given by:

$$f_W(v) = \left(\frac{k}{c}\right)\left(\frac{v}{c}\right)^{k-1} \exp\left[-\left(\frac{v}{c}\right)^k\right]. \tag{2.2}$$

The Rayleigh model is a special and simplified case of the Weibull model. It is obtained when the shape factor $k$ of the Weibull model is assumed to be equal to 2.

The maximum likelihood estimation method is a mathematical expression recognized as a likelihood function of the wind speed data in a time series format. In this method, a lot of numerical iteration can be required to determine the $k$ and $c$ parameters of the Weibull function. The parameter estimation formula of the maximum likelihood method is as follows:

$$k = \left(\frac{\sum_1^n v_i^k \ln(v_i)}{\sum_1^n v_i^k} - \frac{\sum_1^n \ln(v_i)}{n}\right)^{-1}, \tag{2.3}$$

$$c = \left(\frac{1}{n}\sum_1^n v_i^k\right)^{1/k}. \tag{2.4}$$

The average value and standard deviation of the wind speed can be obtained from the following formulas:

$$v_m = \frac{1}{n}\left[\sum_{i=1}^n v_i\right], \tag{2.5}$$

$$\sigma = \left[\frac{1}{n-1}\sum_{i=1}^n (v_i - v_m)^2\right]^{1/2}, \tag{2.6}$$

respectively.

Alternatively, the mean wind speed can be determined from:

$$v_m = \int_0^\infty v f(v)dv, \tag{2.7}$$

if the probability density function is known.

If Eq. (2.7) is solved together with Eq. (2.2) making the substitution of $\xi = (v/c)^k$ for $v$, the following is obtained for the mean wind speed:

$$v_m = c\Gamma\left(1+\frac{1}{k}\right). \tag{2.8}$$

Note that the gamma function has the properties of $\Gamma(x) = \int_0^\infty \xi^{x-1}\exp(-\xi)d\xi$ and $\Gamma(1+x) = x\Gamma(x)$.





### 2.2.2 Power density

The mean power density for the kernel smoothing function becomes:

$$P_K = \sum_{i=1}^{n} \left[ \frac{1}{2} \rho v_{m,i}^3 f_h(v_i) \right].$$

(2.9)

The mean power density for the Weibull function becomes:

$$P_W = \frac{1}{2} \rho c^3 \Gamma \left[ 1 + \frac{3}{k} \right].$$

(2.10)

The mean power density for the Rayleigh model is found to be:


$$P_R = \frac{3}{\pi} \rho v_m^3,$$

(2.11)

where $\rho$ is the air density.

### 2.2.3 Weibull parameters

The relationship between scale factor $c$ and height can be expressed as follows:

$$c / c_{10} = (z / 10)^\alpha.$$

(2.12)

Here $c_{10}$ represents the scale factor at 10-m height, $z$ represents the height, and $\alpha$ represents the power exponent parameter to be estimated.

The relationship between scale factor $k$ and height can be expressed as follows:

$$k = a(z/10)^2 + b'(z/10) + d,$$

(2.13)

where $a$, $b'$, and $d$ are unknown parameters to be fitted to the quadratic function.

In addition, as shown in Figure 6c below, (Justus and Mikhail, 1976) gave the following formula for the shape factor $k$ with height:

$$k_{10} / k = 1 + b_{10} \ln(z/10),$$

(2.14)

where $k_{10}$ is the shape factor at a reference height of 10 m. At a reference height of 10 m, $b = b_{10}$ is just some constant, whose value can be determined by a least squares fit of relation (2.14) to the data.

### 2.2.4 Surface roughness length

When the wind speed at three or more heights is measured, the roughness length calculated by the least square regression (Archer and Jacobson, 2003; Archer, 2005; Golbazi and Archer, 2019) is:





$$\ln(z_0) = \frac{U(z_R)\left\{\sum\left[\ln(z_i)\right]^2 - \ln(z_R)\sum\ln(z_i)\right\} - \ln(z_R)\sum\left[U_i\ln\left(\frac{z_i}{z_R}\right)\right]}{U(z_R)\sum\ln(z_i) - \sum\left[U_i\ln\left(\frac{z_i}{z_R}\right)\right] - N\cdot U(z_R)\ln(z_R)}, \qquad (2.15)$$

where $z_R$ is the reference height, $z_i$ is the height of the other three layers, $N = 4$ which represents 4 vertical layers, and $U_i$ is the
wind speed corresponding to the height of the other three layers. In most cases, it is a purely mathematical statistical method, so this simple mathematical method does not require a physical explanation for roughness estimation.

In addition, the above-mentioned method is obtained from the logarithmic wind speed profile, which is a typical form of wind speed profile under neutral stratification. A calculation of the wind speed at other altitudes under the reference altitude can be obtained from the following formula (Golbazi and Archer, 2019; Archer and Jacobson, 2003):

$$U(z) = U(z_R)\frac{\log\left(\frac{z}{z_0}\right)}{\log\left(\frac{z_R}{z_0}\right)}, \qquad (2.16)$$

where $z_0$ is the estimated surface roughness length, assuming that the friction speed near the ground does not change with height.

## 3 Results and discussion

### 3.1 Comparisons of kernel, Weibull, and Rayleigh models

There are various statistical distribution functions for describing and analyzing wind data, including normal, lognormal, Rayleigh, and Weibull probability distributions (Fagbenle et al., 2011; Ozerdem and Turkeli, 2003). It has been found that the Weibull and Rayleigh distributions are the most accurate and adequate in wind analysis and in interpreting the actual wind speed data and in predicting the characteristics of the prevailing wind profile. A kernel distribution is a nonparametric representation of the PDF of a random variable. A kernel distribution is defined by a smoothing function and a bandwidth
value, which control the smoothness of the resulting density curve (Kafadar et al., 1999). In fact, some scholars have used the probability density distribution of wind speed to compare the advantages and disadvantages of the Weibull distribution and the Rayleigh distribution (Celik, 2004). However, in the present study, by quoting the kernel function distribution close to the actual distribution as a reference, two specified distribution functions are being compared with the kernel function to find out which one can better predict the wind speed data in the area.

The monthly, seasonal, and annual average wind speed values and standard deviations calculated using Eqs. (2.5) and (2.6) for the available time series data are shown in Table 1. It can be seen from Table 1 that the highest average wind speeds occurred in May and December 2019 and in May 2020, and the lowest average wind speeds occurred in February and August 2019. Over about two years, it was found that the average wind speed in the spring of 2019 and 2020 was higher, and the average wind speed in the summer of 2019 and 2020 was lower. During the entire experimental period, the average wind speed





values at 10 m, 30 m, 50 m, and 70 m were 6.02 m/s, 6.78 m/s, 7.25 m/s, and 7.61 m/s, respectively, which also shows that the wind speed value increases with an increase in altitude.

Figure 4 shows the frequency density histogram of the wind speed at 70 m for about two years and the probability density curves of the Weibull, kernel, and Rayleigh distributions. First of all, it is obvious from the frequency histogram that the wind speed at 70 m fluctuated drastically in the autumn of 2018, spring of 2019, and summer of 2020. This conclusion can also be

well explained from Table 1. The shape factor $k$ values of these three specific seasons are 2.18, 2.13, and 2.11 respectively, which are slightly higher than the shape factor $k$ value of the Rayleigh distribution. In combination with Table 1, it is also found that the higher the value of the scale factor $c$, the smoother the three specific probability distribution curves. In contrast, as shown in Figure 4d, its three specific probability density curves are very sharp. Finally, when these three specific probability density curves are fitted to the original wind speed data, the kernel distribution fitting the original wind speed data is not only

the change trend or probability density estimate, but it is closer to the actual frequency distribution.

Although the kernel distribution also has specific parameters to control its probability density curve, it does not have the general form of wind speed distribution. Moreover, the $k$ value of the Weibull distribution is ~2. To select the specific wind speed distribution form suitable for the Ningyuanbailiutu site, therefore, the model prediction accuracies of the Weibull distribution and the Rayleigh distribution for average wind power need to be further compared.

In the present study, the differences between the kernel distribution, Weibull distribution, and Rayleigh distribution are explored when calculating the average wind power density and the frequency distribution using the original wind speed data. The root mean square errors (RMSEs) of the kernel distribution, Weibull distribution, and Rayleigh distribution are 45.75 $W/m^2$, 60.53 $W/m^2$, and 875.34 $W/m^2$, respectively. The Weibull and kernel models return smaller error values in calculating the mean power density compared to the Rayleigh model. The mean power density is estimated by the Rayleigh model to have

a very large absolute error value of 83.12 $W/m^2$ in December 2019. On the other hand, the highest absolute error value occurs in May 2019 with 21.28 $W/m^2$ for the Weibull model. The two-year average absolute percentage error values in calculating the mean power density using the kernel, Weibull, and Rayleigh functions are 1.17%, 1.05%, and 4.20%, respectively.

The mean power densities calculated from the measured probability density distributions and those obtained from the models are shown in Figure 5. The mean power density shows significant monthly and seasonal variation. The minimum average

power density appeared in August 2019 and was only 214.92 $W/m^2$. In addition, smaller mean wind power densities appeared in July and September 2019 and January, July, and August 2020, which were generally lower than 350 $W/m^2$. Generally, the maximum value of monthly mean wind power density reached 862.39 $W/m^2$ in May 2019, and the seasonal mean wind power density peaked in spring 2020.

Although the mean wind power density calculated in this study is in good agreement with the actual grid-connected average

power density assumed (Figure 5a), there is significant difference between the two values. This is because the wind turbines are not connected to the grid due to failures, or other wind turbines are not within the range of the wind measurement tower. As a result, the wind measured by a single wind tower will underestimate the wind speed of other wind turbines.



Analysis of residual error and average percentage error suggests that the average wind power density estimated by the Weibull distribution with specific parameter control is very similar to the kernel distribution, which is closest to the original wind frequency distribution (Figure 5c). The lower limit of the 95% prediction interval is each predicted value minus 1.96 standard deviations, and the upper limit is each predicted value plus 1.96 standard deviations (Figures 5b–d). This suggests that the interval applicability of the three specific distribution models is good. In general, we found that Weibull distribution is applicable for checking the wind speed distribution of the Ningyuanbailiutu site.

**3.2 Vertical characteristics of Weibull parameters**

Figure 6 shows the characteristic variation of the scale factor $c$ and the shape factor $k$ with height estimated from the Weibull distribution for original wind speed data during the study period, exhibiting power exponential and quadratic function variations, respectively.

Table 2 gives in detail the values of $\alpha$, $a$, $b'$, $d$, and $b_{10}$ obtained by the least squares fitting method for each month, each season, and all time periods, and the corresponding RMSEs obtained from the formula. When using the power exponent formula (2.12) to fit the relationship between the scale factor $c$ and the height, the RMSE has the smallest values in January 2019 and July and August 2020. However, in December 2019, January 2020, and February 2020 it has the largest values. This shows that formula (2.12) has a better fitting effect in the winter of 2018, and a poor fitting effect in the winter of 2019. (Justus and Mikhail, 1976) found that the mean value of $\alpha$ was 0.23. In the present study, the mean value of $a$ for each month over the two whole years is 0.117, and the corresponding standard deviation is 0.016.

Figures 6b and 6c indicate that the two different formula forms have a good fitting relationship for shape factor $k$ and height. The RMSEs of Table 2 also suggest that the effect of the quadratic function fitting is better than the logarithmic reciprocal function of (Justus and Mikhail, 1976). The RMSE of the quadratic function fitted to all data for two whole years is 0.0078, but the RMSE of the logarithmic reciprocal function is 0.0214, which is close to a multiple of 1:3. Both these two types of formula are basically applicable only to heights below 100 m. In addition, from a comparison of Figures 6b and 6c, it can be seen that there will be some different trends in the change in the $k$ value with height, and the increasing or decreasing speed of the $k$ value in the form of a quadratic function will be higher than that found by (Justus and Mikhail, 1976) when the height is greater than 70 m. This different trend will lead to large errors in estimating wind energy resources above 70 m. Therefore, it is necessary to intensively observe wind speeds above 70 m and below 100 m in future research, in order to establish a specific function of $k$ varying with height applicable to the site.

**3.3 Spatial–temporal variations in surface roughness length**

The shape of the wind profile is greatly affected by the surface roughness in the direction of the incoming flow. Thus, surface roughness is a key element in wind energy resource evaluation and forecasting models. In calculating aerodynamic roughness,



especially in practical applications, the least squares approximation of the logarithmic profile equation to the measured wind speed profile method has been widely used, referred to as the logarithmic profile method.

After calculating the 15-minute continuous wind speed data using the above method, quality control of the data is carried out. In this study, we have eliminated wind speeds greater than or equal to 6 m/s at 50 m, and the estimated abnormal roughness data is infinitely large or infinitely small. Figure 7 shows that both the average and median monthly roughness length in January, February, and March 2019 are significantly less than those in August, September, and October 2019. The largest value of median roughness was close to 0.19 m in October 2019, and the maximum value of average roughness was approximately

equal to 0.27 m. In June 2020, the median and average roughness values reached 0.18 m and 0.25 m, respectively. The minimum value of median roughness was about 0.10 m in January 2019, and the smallest value of average roughness value was about 0.20 m in January 2020.

In addition, the median and average roughness length were lowest at about 0.12 m and 0.22 m in the winter of 2018 and 2019, while the highest were about 0.15 m and 0.25 m in the autumn of 2019. It is notable that the roughness length steadily increases

from winter to autumn. In short, this suggests that the grassland vegetation in the site area has an obvious wilting period, growing season, and lush period. Compared with the average roughness length, the representative roughness length of the area fitted the median value more closely.

According to the Davenport land type roughness classification (Davenport et al., 2000) and summary of roughness length over the wind-tower sites and the corresponding types (Li et al., 2021), in the case of land types with less vegetation and cropland,

the roughness length is generally estimated to be a slightly rough open area of about 0.10 m. The area we studied belongs to the grassland vegetation type, and the roughness estimate should be around 0.13 m, and it will not be classified as rough; that is, the roughness length is as high as 0.25 m. In addition, in a study (Golbazi and Archer, 2019) on the estimation of sea surface roughness length in coastal waters, it is mentioned that the statistical method uses a single constant value of $z_0$ in the representative area, and the median value can be worth recommending.

Figure 8 shows the variation in the estimated roughness length in 12 different incoming wind directions. When the wind direction is 120° or 240°, the estimated roughness length is highest, and the median value and average value are about 0.23 m and 0.30 m, respectively. Secondly, when the wind direction is 30° or 300°, the estimated roughness length is lowest, and the median value and average value are about 0.08 m and 0.18 m, respectively. Therefore, between the highest and lowest estimated roughness lengths, there is a specific trend of increasing or decreasing. The above phenomenon can be explained in

conjunction with Figure 1 and Figure 3. There is a hillside to the west of the wind tower. Therefore, when the incoming wind direction is 120° or 240°, it is on the windward side or leeward side, respectively, of the wind measuring tower. In this way, there will be a pressure difference, which will increase friction loss and increase the estimate of the effective roughness length. When the incoming wind direction is 30° or 330°, it is found that the wind passing through the wind measurement tower will not be greatly affected by the terrain. The terrain is relatively flat, and the estimated roughness length is close to the normal

value of 0.10 m. In addition, in the plot of roughness length estimation with wind direction, there are obviously more data



points in the wind directions from 180 ° to 330 ° than in the other wind directions. The 240 ° wind direction has the most data points, which also shows that the site has a southwesterly wind blowing all year round.

## 3.4 Extrapolation of the average wind power density

With the scale factor $c$ changing with height in the form of a power function, and shape factor k changing with height in the
form of a quadratic function, the scale factor c and the shape factor k at 75 m, 80 m, 85 m, 90 m, 95 m, and 100 m are calculated. Then the average wind power density (Figure 9b) is calculated for each month, each season, and the whole time period from formula (2.10). On the other hand, when studying the roughness length parameter in the previous section, we assume that the roughness length calculated from the four-layer height is constant. Then through the logarithmic form of formula (2.16), we can calculate the wind speed values at 75 m, 80 m, 85 m, 90 m, 95 m, and 100 m every 15 minutes. Finally, the "reference
average power density" (Figure 9a) at six specific heights can also be obtained.

Both the power-law model and the logarithmic model can estimate the average wind power density of six specific heights, and it can be found from Figure 9 that the values estimated by the two methods show greater differences in autumn and winter, and smaller differences in spring and summer. In addition, the two different models both show that the average power density values are largest in spring and smallest in summer. Although the average power density values increased with height over the
whole experiment, the gradient of the increase in average power density values with height is largest in autumn and winter, and smallest in spring and summer. Generally speaking, the difference between the estimated average power density values is very small. However, the data and methods used in the estimation of the two models are different. The result of this estimation gives us important guidance for studying two Weibull parameters, namely the scale factor $c$ and the shape factor $k$, and the surface roughness length parameter.

## 4. Conclusions

The present work investigated the scale factor $c$ and the shape factor $k$ that affect the Weibull distribution of wind speed, by directly estimating the energy potential of the wind speed resource at four different heights, and the surface roughness length parameter which directly affects the shape and law of the wind profile. The main conclusions are given as follows:

The two-year average absolute percentage error values in calculating the mean power density using the kernel, Weibull, and
Rayleigh functions are 1.17%, 1.05%, and 4.20%, respectively. The Weibull wind speed distribution model is the most suitable wind speed distribution model for the Ningyuanbailiutu site. The scale factor $c$ increases with an increase in height, showing an obvious form of power function. The shape factor $k$ increases or decreases with height and has two different forms, which are the reciprocal of the quadratic function and the logarithmic function. For the further determination of the changes in form factor with height, in the future, it will be necessary to conduct intensive observations for heights above 70 m and below 100
m.





When estimating the surface roughness length, the median value is selected as the representative value of the surface roughness length. This is based not only on recognition of actual previous research, but also on confirmation of actual grassland vegetation types. Although the statistically calculated $z_0$ does not have a proper physical explanation, it gives the most accurate wind speed estimate at the required height. The estimated roughness length varies with the seasons of the grassland vegetation at

the site. The estimated roughness lengths of the wilting period, growing season, and lush period are about 0.12 m, 0.13 m, and 0.15 m, respectively. The estimated surface roughness length will be affected by the windward and leeward sides. When the wind flows across the hillside, there will be a pressure difference, which will increase the friction loss and increase the estimated effective roughness length. The prevailing wind direction at this site is 240 °, which happens to be the direction of the windward side of the site. The estimated roughness length is about 0.23 m. Finally, the power-law model and the

logarithmic model were employed to estimate the average power density values at 75 m, 80 m, 85 m, 90 m, 95 m, and 100 m. The two models show greater differences in autumn and winter, and smaller differences in spring and summer. The gradient of the increase in average power density values with height is largest in autumn and winter, and smallest in spring and summer. In general, under a carbon-neutral background, the determination of the potential for economical and clean wind energy resources is an important scientific issue in the development of renewable energy worldwide. Our research has determined the

possible relationship between Weibull natural wind mesoscale parameter $c$ and shape factor $k$ with height under the conditions of a desert steppe terrain in northern China, which has great potential in wind power generation, but there is a lack of comprehensive investigations into key parameters for estimating wind power density from tower data. In the present study, we have gained an enhanced understanding of the seasonal changes in the surface roughness of the desert grassland and the changes in the incoming wind direction. Our findings also have important implications for the assessment of wind energy

resources for the establishment of new wind farms in areas experiencing varied desert steppe terrains throughout the world.



**Code availability**

The model and all implementation and analysis codes in this paper are based on MatLab and python, which can be available upon request to the author (20191203039@nuist.edu.cn).

**Data Availability**

 The data in this paper comes from an observation wind tower, which can be available upon request from the author (20191203039@nuist.edu.cn).

**Author contributions**

YY and ZG were responsible for conceptualization, supervision and funding acquisition. SZ developed the software and
prepared the original draft. SZ and YY developed the methodology and carried out formal analysis. XX and SZ validated data. ZG,YY, XX, ZD, and YL were reviewed and edited the text. SZ was responsible for visualization. All authors have read and agreed to the published version of the paper.

**Competing interests**

The authors declare that they have no conflict of interest.






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





**Table 1: Calculated monthly, seasonal, and annual distribution parameters based on the time series wind speed data measured every 15 minutes from Damaoqi Wind Tower.**

| | 10 m | | | | 30 m | | | | 50 m | | | | 70 m | | | |
|---|---|---|---|---|---|---|---|---|---|---|---|---|---|---|---|---|
| | $v_m$ | $\sigma$ | $c$ | $k$ | $v_m$ | $\sigma$ | $c$ | $k$ | $v_m$ | $\sigma$ | $c$ | $k$ | $v_m$ | $\sigma$ | $c$ | $k$ |
| Sep 2018 | 6.34 | 3.13 | 7.14 | 2.09 | 7.15 | 3.48 | 8.02 | 2.09 | 7.65 | 3.55 | 8.60 | 2.23 | 8.05 | 3.67 | 9.06 | 2.30 |
| Oct 2018 | 5.84 | 3.41 | 6.56 | 1.77 | 6.66 | 3.77 | 7.47 | 1.81 | 7.16 | 3.89 | 8.06 | 1.91 | 7.52 | 4.04 | 8.48 | 1.95 |
| Nov 2018 | 6.32 | 2.80 | 7.11 | 2.35 | 7.25 | 3.22 | 8.14 | 2.33 | 7.83 | 3.39 | 8.79 | 2.40 | 8.27 | 3.65 | 9.29 | 2.36 |
| Dec 2018 | 6.55 | 3.01 | 7.36 | 2.25 | 7.48 | 3.37 | 8.40 | 2.31 | 8.02 | 3.58 | 9.00 | 2.34 | 8.41 | 3.82 | 9.43 | 2.28 |
| Jan 2019 | 6.14 | 2.66 | 6.89 | 2.40 | 6.96 | 2.96 | 7.81 | 2.45 | 7.40 | 3.13 | 8.31 | 2.49 | 7.68 | 3.35 | 8.62 | 2.40 |
| Feb 2019 | 5.05 | 3.57 | 5.57 | 1.45 | 5.48 | 3.98 | 6.00 | 1.39 | 5.73 | 4.20 | 6.28 | 1.39 | 5.95 | 4.37 | 6.54 | 1.40 |
| Mar 2019 | 6.38 | 3.03 | 7.16 | 2.16 | 7.20 | 3.42 | 8.08 | 2.17 | 7.66 | 3.59 | 8.61 | 2.21 | 8.02 | 3.72 | 9.03 | 2.27 |
| Apr 2019 | 6.42 | 3.43 | 7.19 | 1.89 | 7.19 | 3.85 | 8.06 | 1.88 | 7.78 | 4.05 | 8.75 | 1.98 | 8.17 | 4.22 | 9.22 | 2.02 |
| May 2019 | 7.19 | 3.81 | 8.12 | 1.97 | 8.08 | 4.10 | 9.11 | 2.05 | 8.65 | 4.26 | 9.76 | 2.12 | 9.02 | 4.39 | 10.20 | 2.17 |
| Jun 2019 | 5.94 | 2.87 | 6.70 | 2.17 | 6.68 | 3.14 | 7.53 | 2.23 | 7.19 | 3.30 | 8.10 | 2.29 | 7.55 | 3.45 | 8.52 | 2.32 |
| Jul 2019 | 5.00 | 2.84 | 5.60 | 1.78 | 5.57 | 3.25 | 6.21 | 1.70 | 6.08 | 3.47 | 6.80 | 1.77 | 6.39 | 3.56 | 7.17 | 1.83 |
| Aug 2019 | 4.44 | 2.39 | 4.98 | 1.87 | 4.96 | 2.67 | 5.54 | 1.84 | 5.41 | 2.74 | 6.07 | 2.02 | 5.70 | 2.82 | 6.43 | 2.11 |
| Sep 2019 | 5.26 | 2.40 | 5.90 | 2.25 | 6.06 | 2.78 | 6.78 | 2.24 | 6.58 | 2.91 | 7.39 | 2.37 | 7.00 | 3.02 | 7.87 | 2.47 |
| Oct 2019 | 6.00 | 3.33 | 6.72 | 1.83 | 6.83 | 3.76 | 7.64 | 1.82 | 7.40 | 4.00 | 8.30 | 1.88 | 7.84 | 4.14 | 8.82 | 1.96 |
| Nov 2019 | 6.51 | 3.55 | 7.31 | 1.88 | 7.43 | 3.96 | 8.32 | 1.89 | 7.94 | 4.10 | 8.94 | 2.00 | 8.38 | 4.23 | 9.46 | 2.08 |
| Dec 2019 | 6.89 | 2.84 | 7.72 | 2.56 | 7.82 | 3.30 | 8.75 | 2.48 | 8.46 | 3.49 | 9.48 | 2.59 | 8.98 | 3.67 | 10.07 | 2.64 |
| Jan 2020 | 5.37 | 2.71 | 6.03 | 2.01 | 5.69 | 3.21 | 6.34 | 1.76 | 6.03 | 3.40 | 6.74 | 1.79 | 6.31 | 3.62 | 7.07 | 1.79 |
| Feb 2020 | 6.34 | 3.29 | 7.09 | 1.93 | 7.11 | 3.63 | 7.97 | 1.99 | 7.60 | 3.91 | 8.51 | 1.97 | 8.09 | 4.03 | 9.10 | 2.08 |
| Mar 2020 | 6.65 | 3.21 | 7.48 | 2.14 | 7.60 | 3.49 | 8.54 | 2.26 | 8.10 | 3.65 | 9.12 | 2.34 | 8.56 | 3.74 | 9.64 | 2.44 |
| Apr 2020 | 5.66 | 3.44 | 6.30 | 1.64 | 6.36 | 3.68 | 7.10 | 1.74 | 6.72 | 3.88 | 7.52 | 1.75 | 7.12 | 3.97 | 8.01 | 1.85 |
| May 2020 | 6.88 | 3.82 | 7.72 | 1.84 | 7.63 | 4.10 | 8.56 | 1.90 | 8.06 | 4.25 | 9.05 | 1.94 | 8.39 | 4.36 | 9.44 | 1.99 |
| Jun 2020 | 6.53 | 2.95 | 7.34 | 2.29 | 7.25 | 3.30 | 8.15 | 2.27 | 7.58 | 3.51 | 8.53 | 2.25 | 7.88 | 3.67 | 8.89 | 2.26 |
| Jul 2020 | 5.35 | 2.70 | 6.01 | 2.04 | 6.04 | 2.96 | 6.79 | 2.10 | 6.35 | 3.07 | 7.14 | 2.14 | 6.61 | 3.23 | 7.44 | 2.12 |
| Aug 2020 | 5.35 | 2.74 | 6.01 | 2.00 | 6.09 | 3.06 | 6.83 | 2.03 | 6.43 | 3.21 | 7.21 | 2.04 | 6.69 | 3.35 | 7.51 | 2.04 |
| 2018 autumn | 6.16 | 3.14 | 6.94 | 2.03 | 7.02 | 3.51 | 7.88 | 2.05 | 7.54 | 3.63 | 8.49 | 2.15 | 7.94 | 3.80 | 8.94 | 2.18 |
| 2018 winter | 5.94 | 3.15 | 6.67 | 1.92 | 6.68 | 3.54 | 7.49 | 1.91 | 7.09 | 3.76 | 7.96 | 1.91 | 7.39 | 3.98 | 8.29 | 1.88 |
| 2019 spring | 6.67 | 3.46 | 7.49 | 1.98 | 7.49 | 3.82 | 8.42 | 2.01 | 8.03 | 4.00 | 9.04 | 2.08 | 8.41 | 4.14 | 9.49 | 2.13 |
| 2019 summer | 5.12 | 2.78 | 5.74 | 1.88 | 5.73 | 3.11 | 6.41 | 1.85 | 6.22 | 3.27 | 6.98 | 1.95 | 6.54 | 3.38 | 7.37 | 2.01 |





| | | | | | | | | | | | | | | | | |
|---|---|---|---|---|---|---|---|---|---|---|---|---|---|---|---|---|
| 2019 autumn | 5.92 | 3.18 | 6.65 | 1.90 | 6.77 | 3.58 | 7.59 | 1.91 | 7.31 | 3.75 | 8.22 | 2.00 | 7.74 | 3.88 | 8.73 | 2.08 |
| 2019 winter | 6.20 | 3.02 | 6.95 | 2.10 | 6.87 | 3.49 | 7.69 | 1.98 | 7.36 | 3.74 | 8.25 | 2.00 | 7.79 | 3.93 | 8.76 | 2.04 |
| 2020 spring | 6.41 | 3.54 | 7.18 | 1.84 | 7.20 | 3.81 | 8.08 | 1.92 | 7.64 | 3.98 | 8.58 | 1.96 | 8.03 | 4.08 | 9.05 | 2.04 |
| 2020 summer | 5.73 | 2.85 | 6.45 | 2.07 | 6.45 | 3.16 | 7.25 | 2.10 | 6.78 | 3.31 | 7.62 | 2.11 | 7.05 | 3.47 | 7.94 | 2.11 |
| Whole period | 6.02 | 3.18 | 6.76 | 1.94 | 6.78 | 3.55 | 7.60 | 1.94 | 7.25 | 3.73 | 8.14 | 2.00 | 7.61 | 3.88 | 8.57 | 2.03 |





**Table 2. The values of various parameters in different time periods and the corresponding root mean square errors (RMSEs) under least squares formula fitting.**

| | $\alpha$ | RMSE_c | $a$ | $b'$ | $d$ | RMSE_k | $b_{10}$ | RMSE_k (Justus and Mikhail, 1976) |
|---|---|---|---|---|---|---|---|---|
| Sep 2018 | 0.1180 | 0.0672 | 0.0040 | 0.0065 | 2.0692 | 0.0245 | −0.0372 | 0.0500 |
| Oct 2018 | 0.1293 | 0.0493 | −0.0008 | 0.0382 | 1.7233 | 0.0141 | −0.0434 | 0.0242 |
| Nov 2018 | 0.1339 | 0.058 | −0.0009 | 0.0133 | 2.3262 | 0.0204 | −0.0050 | 0.0216 |
| Dec 2018 | 0.1260 | 0.0296 | −0.0070 | 0.0615 | 2.1956 | 0.0079 | −0.0136 | 0.0268 |
| Jan 2019 | 0.1154 | 0.0095 | −0.0091 | 0.0748 | 2.3257 | 0.0105 | −0.0114 | 0.0371 |
| Feb 2019 | 0.0782 | 0.0468 | 0.0042 | −0.0409 | 1.4844 | 0.0061 | 0.0238 | 0.0136 |
| Mar 2019 | 0.1165 | 0.0404 | 0.0027 | −0.0034 | 2.1569 | 0.0015 | −0.0183 | 0.0226 |
| Apr 2019 | 0.1221 | 0.0972 | 0.0029 | 0.0004 | 1.8797 | 0.0171 | −0.0253 | 0.0340 |
| May 2019 | 0.1148 | 0.0559 | −0.0019 | 0.0483 | 1.9256 | 0.0040 | −0.0432 | 0.0139 |
| Jun 2019 | 0.1195 | 0.0652 | −0.0019 | 0.0399 | 2.1315 | 0.0024 | −0.0312 | 0.0092 |
| Jul 2019 | 0.1207 | 0.1040 | 0.0090 | −0.0608 | 1.8252 | 0.0163 | 0.0011 | 0.0476 |
| Aug 2019 | 0.1245 | 0.0945 | 0.0076 | −0.0171 | 1.8682 | 0.0328 | −0.0409 | 0.0700 |
| Sep 2019 | 0.1432 | 0.0736 | 0.0072 | −0.0185 | 2.256 | 0.0224 | −0.0320 | 0.0607 |
| Oct 2019 | 0.1339 | 0.0928 | 0.0052 | −0.0191 | 1.8402 | 0.0046 | −0.0227 | 0.0356 |
| Nov 2019 | 0.1280 | 0.0667 | 0.0037 | 0.0062 | 1.8609 | 0.0149 | −0.0392 | 0.0414 |
| Dec 2019 | 0.1304 | 0.1029 | 0.0081 | −0.0481 | 2.5904 | 0.0255 | −0.0051 | 0.0525 |
| Jan 2020 | 0.0729 | 0.1152 | 0.0153 | −0.1547 | 2.1364 | 0.0366 | 0.0795 | 0.0503 |
| Feb 2020 | 0.1202 | 0.1037 | 0.0036 | −0.0077 | 1.9484 | 0.0230 | −0.0255 | 0.0329 |
| Mar 2020 | 0.1265 | 0.0519 | −0.0017 | 0.0623 | 2.0806 | 0.0098 | −0.0577 | 0.0193 |
| Apr 2020 | 0.1172 | 0.0726 | 0.0002 | 0.0316 | 1.6145 | 0.0207 | −0.0522 | 0.0248 |
| May 2020 | 0.1006 | 0.0393 | −0.0003 | 0.0260 | 1.8191 | 0.0015 | −0.0327 | 0.0123 |
| Jun 2020 | 0.0963 | 0.0267 | 0.0019 | −0.0222 | 2.3162 | 0.0042 | 0.0099 | 0.0063 |
| Jul 2020 | 0.1092 | 0.0145 | −0.0056 | 0.0601 | 1.9787 | 0.0042 | −0.0260 | 0.0153 |
| Aug 2020 | 0.1143 | 0.0122 | −0.0022 | 0.0245 | 1.9728 | 0.0007 | −0.0125 | 0.0047 |
| 2018 autumn | 0.1269 | 0.0581 | 0.0005 | 0.0235 | 1.9993 | 0.0189 | −0.0307 | 0.0297 |
| 2018 winter | 0.1104 | 0.0247 | −0.0009 | 0.0009 | 1.9214 | 0.0073 | 0.0083 | 0.0095 |
| 2019 spring | 0.1177 | 0.0634 | 0.0008 | 0.0194 | 1.9526 | 0.0070 | −0.0311 | 0.0217 |
| 2019 summer | 0.1218 | 0.0891 | 0.0056 | −0.0210 | 1.8881 | 0.0173 | −0.0215 | 0.0435 |





| | | | | | | | | |
|---|---|---|---|---|---|---|---|---|
| 2019 autumn | 0.1346 | 0.0781 | 0.0046 | −0.0049 | 1.8979 | 0.0122 | −0.0333 | 0.0415 |
| 2019 winter | 0.1108 | 0.1087 | 0.0099 | −0.0867 | 2.1701 | 0.0139 | 0.0265 | 0.0376 |
| 2020 spring | 0.1148 | 0.0542 | −0.0003 | 0.0353 | 1.8061 | 0.0111 | −0.0463 | 0.0183 |
| 2020 summer | 0.1059 | 0.0148 | −0.0023 | 0.0246 | 2.0459 | 0.0005 | −0.0110 | 0.0057 |
| Whole period | 0.1178 | 0.0596 | 0.0020 | 0.0008 | 1.9305 | 0.0078 | −0.0185 | 0.0214 |





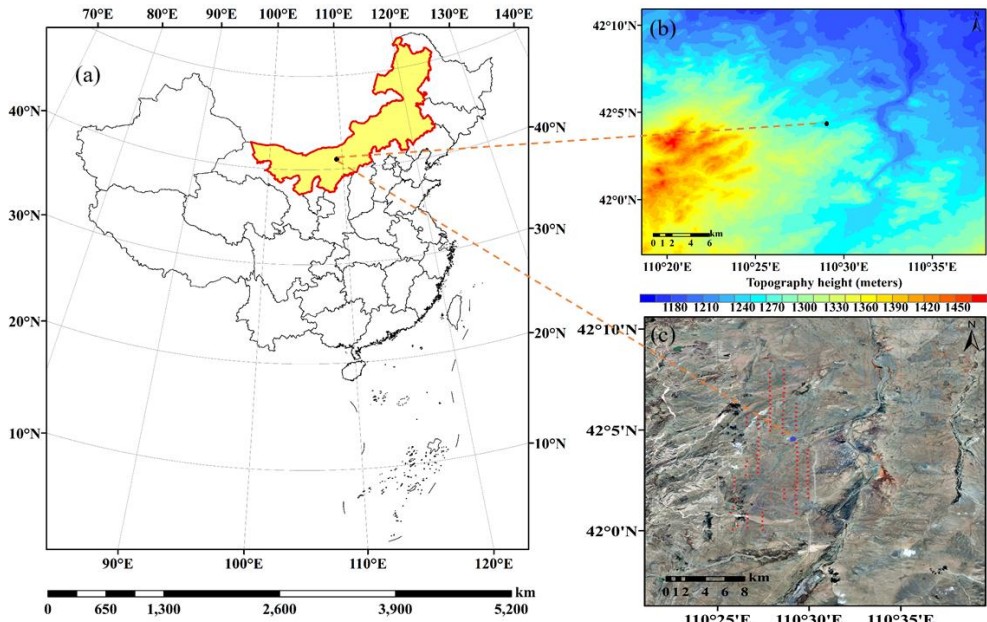

**Figure 1: (a) The observation site marked as a black spot in the Inner Mongolia Autonomous Region of China (the red line indicates**
**the border of the Inner Mongolia Autonomous Region); (b) Terrain elevation map of the 28 km * 28 km grid; (c) Google satellite**
**historical imagery of the 28 km * 28 km grid (from © Google Maps 2021); the red dots indicate wind turbines.**

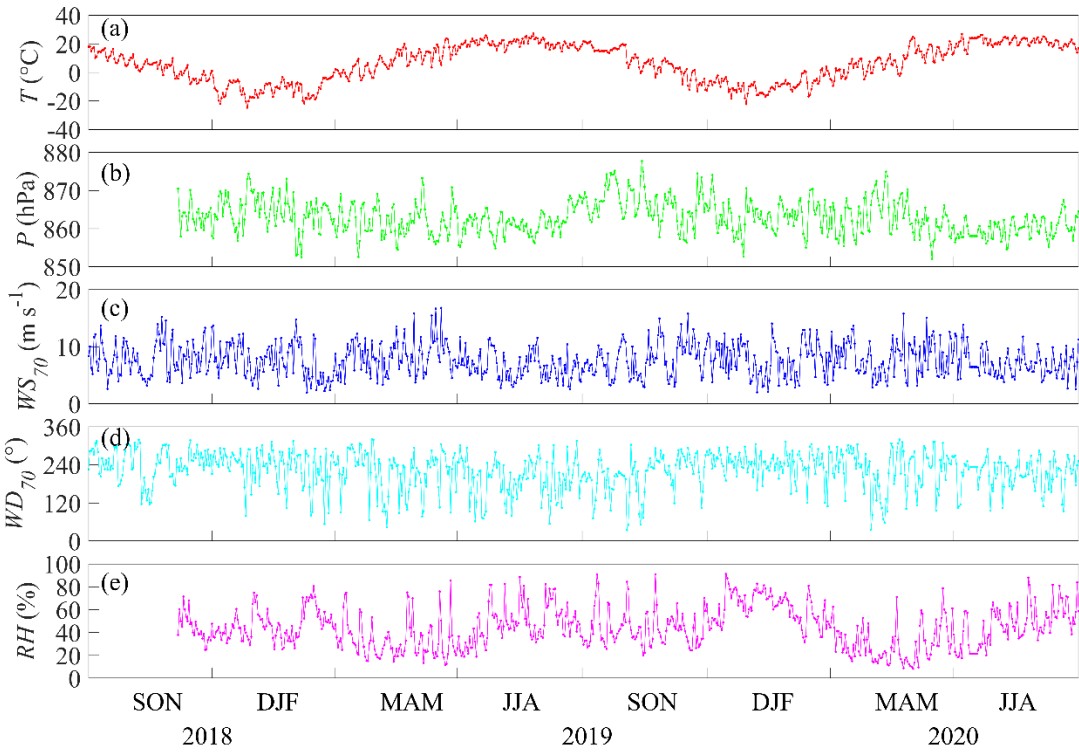

**Figure 2: (a), (b), (c), (d), and (e) represent daily average temperature (℃), daily average pressure (hPa), 70-m daily average wind**
**speed (m/s), 70-m daily average wind direction ( °), and the average daily relative humidity (%), respectively.**

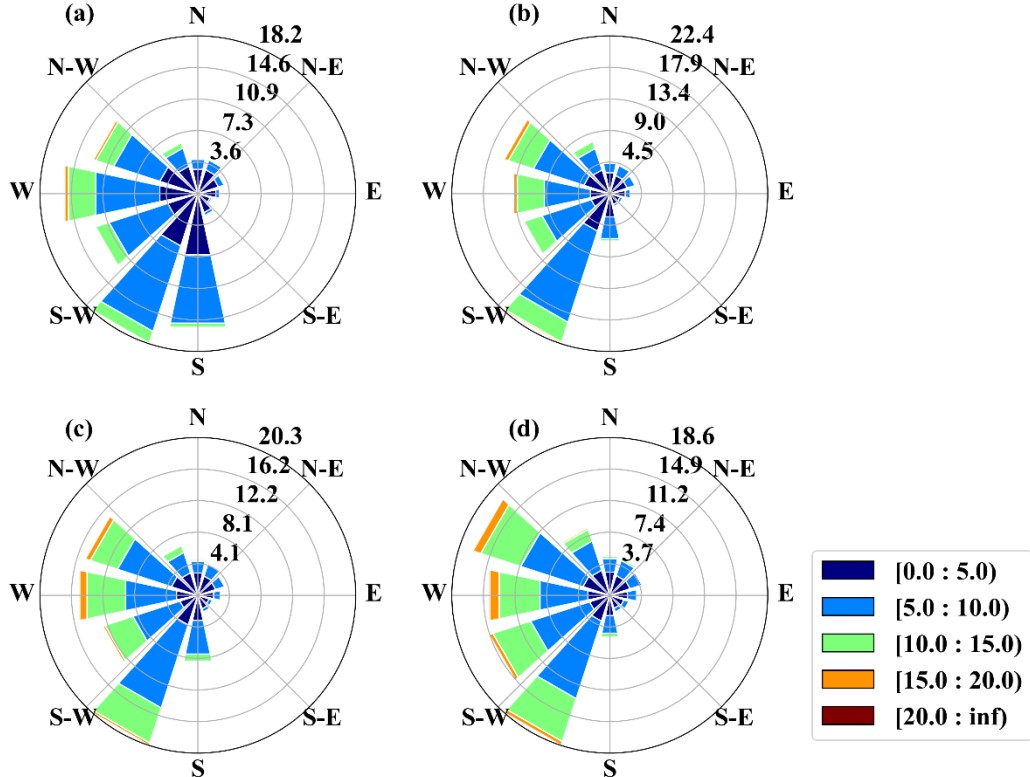

**Figure 3: The distribution of the high-altitude wind speed and direction rose diagram of the wind measurement tower of the Ningyuanbailiutu wind farm from September 2018 to August 2020. (a), (b), (c), and (d) represent 10 m, 30 m, 50 m, and 70 m in altitude, respectively.**






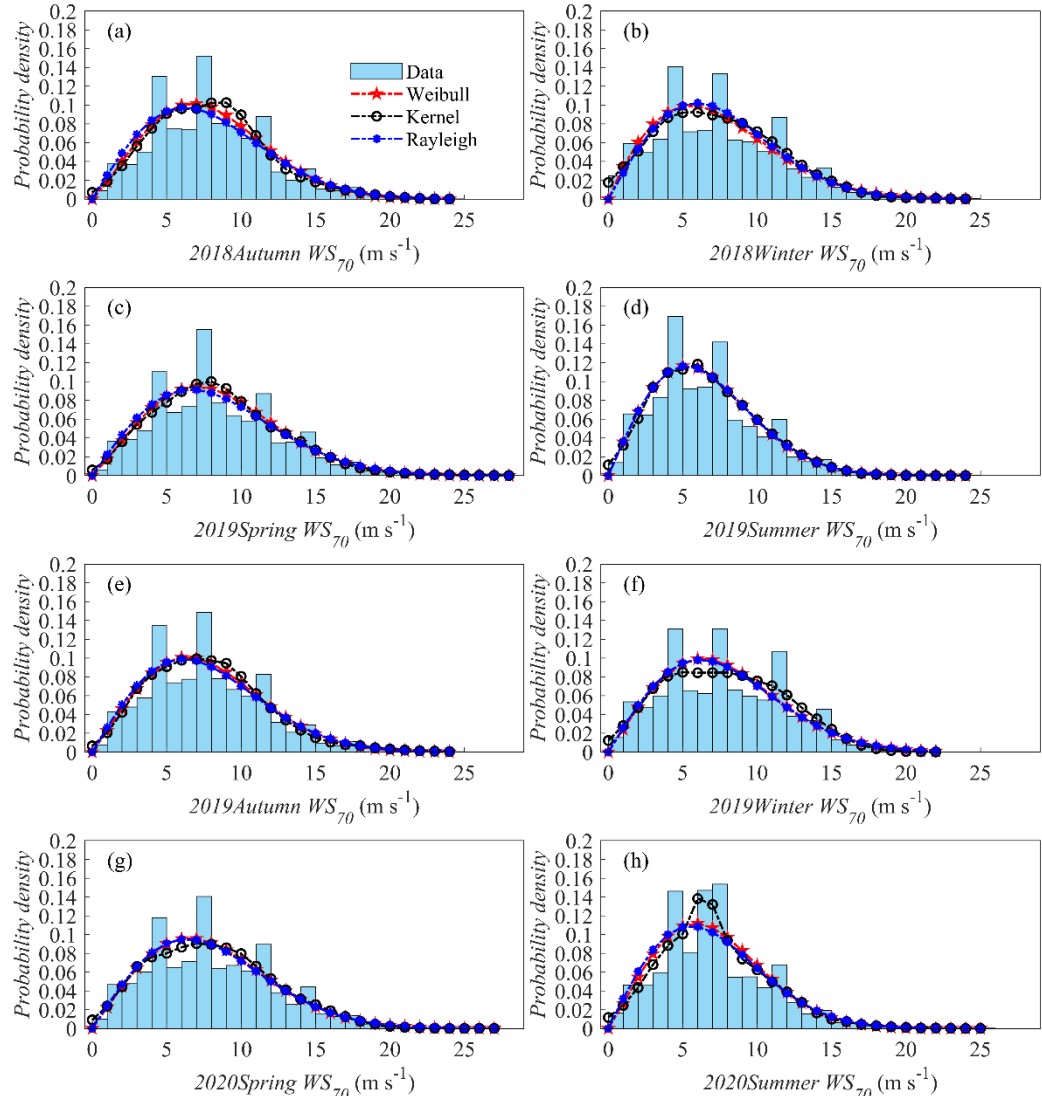

**Figure 4: Frequency density histogram of wind speed at 70-m height from autumn 2018 to summer 2020; the probability density curve obtained by fitting the Weibull, kernel, and Rayleigh distribution functions to the original data. (a), (b), (c), (d), (e), (f), and (h) represent each season, respectively.**





**Figure 5: (a) The average power density values calculated by the four distributions: the frequency distribution of the original wind speed time series data, the kernel distribution, the Weibull distribution, and the Rayleigh distribution, for each month, each season, and the total period of about 2 years. The dotted red line represents the actual grid-connected average power density when the fan blade length is assumed to be 41.5 m. (b), (c), and (d) represent the residual and 95% confidence interval under the three specific probability model distributions, respectively.**






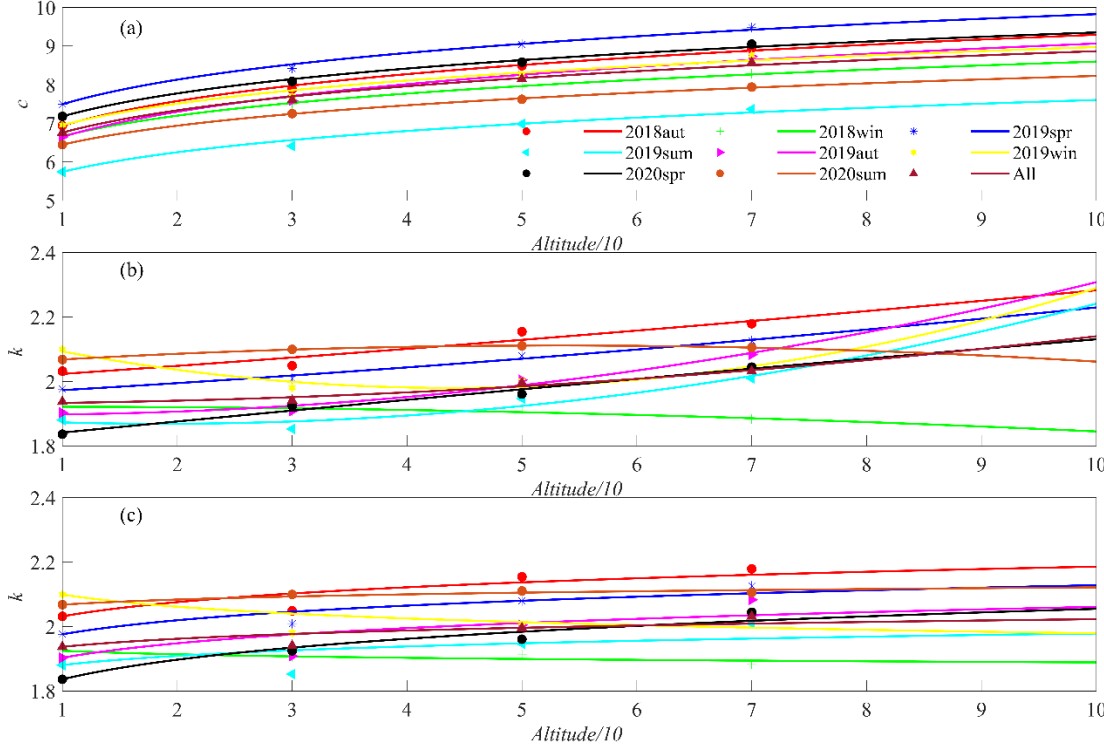


**Figure 6: (a) The characteristic variation of scale factor c with height based on equation (2.12). The characteristic variation of shape factor k with height based on (b) equation (2.13) and (c) equation (2.14).**





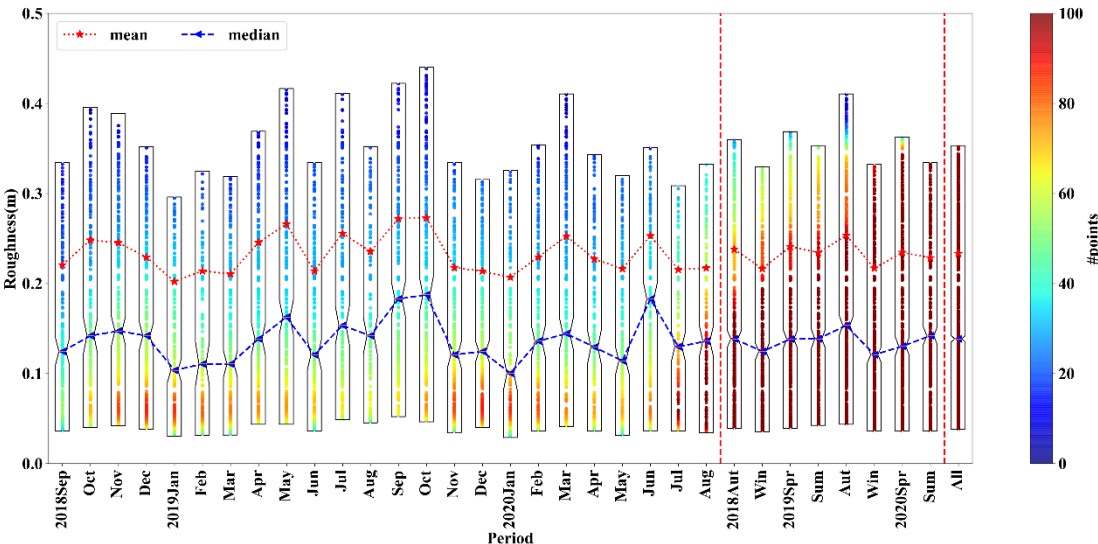

Figure 7: The average and median values of estimated roughness in each month for the total period.





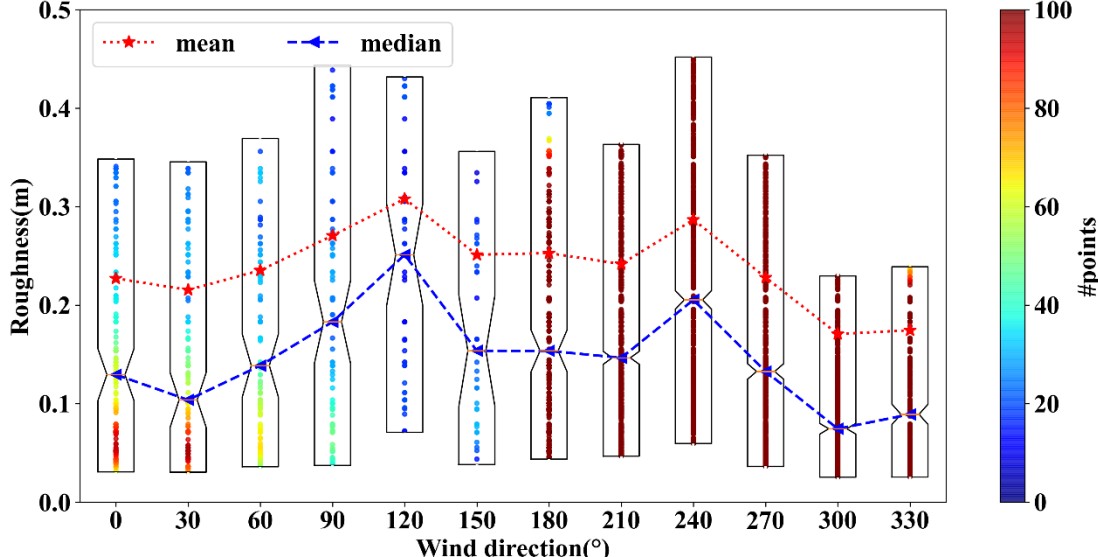

**Figure 8: The average and median values of estimated roughness in 12 different directions of incoming flow.**





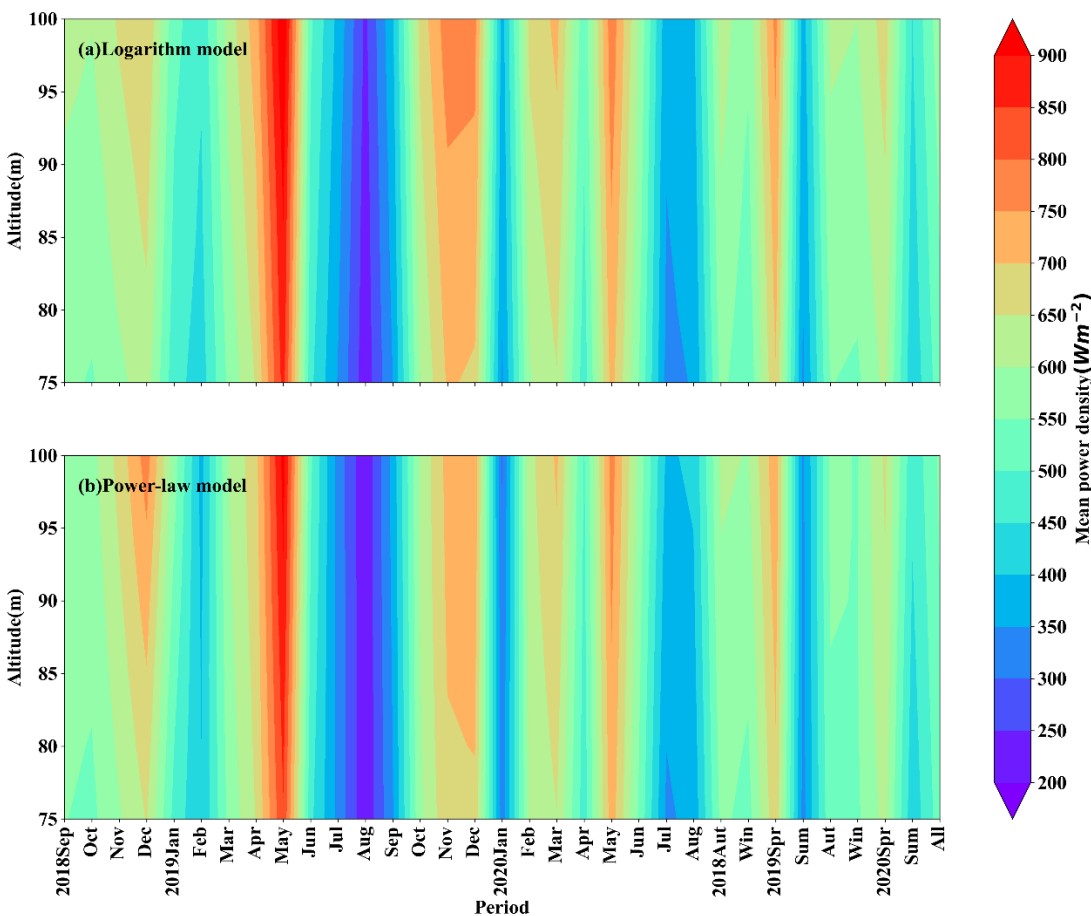


**Figure 9: The logarithmic model (a) and power-law model (b) estimate the average wind energy density at six specific heights (75 m, 80 m, 85 m, 90 m, 95 m, and 100 m) of the wind measurement tower in each month, for every season and over the whole period.**