# Peer review of "Estimating vertical wind power density by using tower observation and empirical models over varied desert steppe terrain in northern China"

_Atmospheric Measurement Techniques, 2021_

## Author Comment (AC1)

**Response to reviewer comments**

We are sincerely grateful to editor and reviewers for their valuable time spent on reviewing our manuscript. The comments are very helpful and valuable, and we have addressed some issues raised by the reviewers in the revised manuscript. Please find our point-by-point response (in blue font) to the comments (in black font) raised by reviewers.

**Reviewer 1**

This study compared three wind speed distributions of kernel, Weibull, and Rayleigh model to estimate vertical average wind power density by using meteorological tower data over varied desert steppe terrain. The topic is very interesting and has important implications for accurate and reliable wind energy evaluation. This study has the potential to provide new insights about three key parameters ($c$, $k$, and $z_0$) should be dynamically considered for estimating wind energy resources under varied desert steppe terrain contexts. The manuscript is written clearly and organized. While I found some minor issues need to be addressed. My recommendation is to accept with minor revision.

**Response:** Many thanks for your positive and valuable comments, and kind suggestions in both quick review and current rounds. We hope our revisions have properly addressed the various concerns and issues raised.

Lines 115, 117, … and figure 2, what are the height for daily average temperature and relative humidity? please clarify them in the main text and figure captions.

**Response:** Many thanks for your question and kind suggestion. The height of the daily average temperature and relative humidity is 2 m. Amended.

[Figure]

Why did you give the shading in tables 1 and 2, any information or implications?

**Response:** Many thanks for your question and kind suggestion. Because table 1 and table 2 are relatively long, the shades of red and blue can be used to represent the size of the value. The red shading and blue shading represent the larger and smaller values in the table, respectively. The darker the color, the more extreme the value. It is convenient to compare with Figure 4 and Figure 6, for example, the size of the scale factor c in Figure 4 represents the approximate position of the probability density maximum value of the distribution curve.

Please accordingly revise all the x-axis titles, for instance in figure 4a, it is should be that WS70 (ms-1) during Autumn in 2018.
**Response:** Many thanks for your kind suggestion. We have revised all the x-axis titles.

[Figure]

What is the unit of legend in figure 5?
**Response:** Many thanks for your question. The unit of the legend is W/m$^2$. We have added legend unit in the caption of Figure 5.

It is unclear for right y x-axis title, confused.

**Response:** Many thanks for your question. The total average power density on the right y-axis in (a) is the average power density of the grid-connected wind turbine with a radius of 41.5m represented by the red dots in Figure 1c. We have added annotation of right y x-axis title to the caption of Figure 5.

Lines 193-194: by quoting the kernel function distribution close to the actual distribution as a reference, two specified distribution functions are being compared with the kernel function to find out which one can better predict the wind speed data in the area. This sentence should be rephrased, for example of "---close to the actual distribution—" , can you give some comparison results with actual measurements?

**Response:** Thank you very much for the constructive suggestions. We have modified the sentence as following:

In our present work, the kernel function exhibits the feature of the smooth function, and also is closer to the actual frequency distribution (Figure 4), which can be used to fit the original wind speed data. Therefore, the kernel function is employed as a medium to compare the pros and cons of the Weibull and Rayleigh functions in the desert steppe area. The above part has been added in Lines 308–311 in the revised manuscript.

In addition, we also give some comparison results with actual measurements at lines 209－213. The differences between the kernel distribution, Weibull distribution, and Rayleigh distribution are explored when calculating the average wind power density and the frequency distribution using the original wind speed data. The two-year average absolute percentage error values in calculating the mean power density using the kernel, Weibull, and Rayleigh functions are 1.17%, 1.05%, and 4.20%, respectively. The root mean square errors (RMSEs) of the kernel distribution, Weibull distribution, and Rayleigh distribution are 45.8 $W/m^2$, 60.5 $W/m^2$, and 875.3$W/m^2$, respectively.

An individual discussion section should be strengthen and added, especially for comparing the present result with others, and uncertainty for your results. Please reorganize the present section 3 and extract discussion parts, for example, Lines 185-194 should be moved to discussion section.

**Response:** Many thanks for your kind suggestion. We extracted a part from Section 3 in the original manuscript and added Section 4 of Discussion part in the revised manuscript, and added Table 3 as a description of the Discussion part.

**Table 3.** Review of scale factor *c*, shape factor *k*, surface roughness $z_0$, and yearly mean absolute percentage error (MAPE) over different topography.

| Topography and Climate context | Location | Period | *c* (m/s) | *k* | $z_0$ (m) | yearly MAPE(%) | Reference |
|---|---|---|---|---|---|---|---|
| 1. A plain area mostly, with slopes rising 50–100 m. 2. Temperate oceanic context | Cardiff, Wales (51.30°N, 3.13°W) | 1991 | 3.25 | 1.79 | − | 3.60 (Weibull) | |
| | | 1994 | 3.16 | 1.76 | | 2.72 (Weibull) | |
| | | 1995 | 2.84 | 1.75 | | 1.97 (Weibull) | |
| | | 1996 | 2.71 | 1.64 | | 2.25 (Weibull) | |
| 1. Lowland of undulating hills, including the floodplains mostly below 600 m. Forested mountain slopes rising to 1200 m. Upland of steep ridges, mountain peaks. 2. Subtropical monsoon humid context | Canberra, Australia (35.18°S, 149.08°E) | | 2.33 | 1.24 | | 4.11 (Weibull) | (Celik, 2003) |
| 1. Mean Altitude with 1560 m above sea level 2. Temperate oceanic context | Davos, Switzerland (46.48°N, 9.50°E) | − | 2.53 | 1.30 | | 4.73 (Weibull) | |
| 1. Altitude with 50 m above sea level. 2. Subtropical Mediterranean context | Athens, Australia (38.00°N, 23.44°E) | | 2.79 | 1.40 | | 1.57 (Weibull) | |
| 1. Altitude with 850 m above sea level. 2. High Anatolian Plateau. 3. Temperate continental context | Ankara, Turkey (39.55°N, 32.50°E) | | 2.65 | 1.60 | | 1.35 (Weibull) | |
| 1. Mediterranean Sea coast 2. Subtropical Mediterranean context | Iskenderun, Turkey (36.35°N, 36.10°E) | 1996 | 2.62 | 1.43 | | 4.90 (Weibull) 36.50 (Rayleigh ) | (Celik, 2004) |
| 1. At 10-m height. 2. Subtropical Mediterranean context | Canakkale, Turkey (40.14°N, 26.42°E) | 2000–2005 | 7.20 | 1.80 | | 7.30 (Weibull) 13.00 (Rayleigh ) | (Celik, 2011) |
| 1. Approximately 100 km east of Tehran city. (At 10-m height). 2. Continental semi-arid context | Firouzkooh, Iran (35.72°N, | 2001–2010 | 6.47 | 2.61 | ~0.14 | 55.00 (Weibull) 55.00 | (Pishgar-Komleh et al., |

| | | | | | | |
|---|---|---|---|---|---|---|
| | 52.40°E) | | | | (Rayleigh ) | 2015) |
| 1. Typical desert grassland. (at 70-m height).
 2. Middle temperate zone and semi-arid continental context | Inner Mongolia, China (42.07°N, 110.48°E) | Sep 2018– Aug 2020 | 9.49 | 2.13 | ~0.13 | 1.05 (Weibull) 4.20 (Rayleigh ) | This study |

Please supply the different results between figure 9a and 9b for better showing the differences at different seasonal and heights, and then discuss their potential cause.

**Response:** Thanks for your kind suggestion. We have added a Figure 9c to illustrate the difference between Figure 9a and Figure 9b, and provided the results of the difference in lines 293–295 of the revised manuscript. Figure 9c shows that relative to the power-law model, the average power density of the logarithmic model extrapolated at 70–100 m is smaller in the winter of 2018 and in July and August of 2020, while larger in other experimental periods.

We discuss the reasons for this discrepancy at lines 346–348 in Section 4.2 of the Discussion section in the revised manuscript. The reason for the difference is that the shape factor $k$ in the winter of 2018 and in July and August of 2020 shows a decreasing trend with height, and the average wind energy density is inversely proportional to the shape factor $k$, according to formula (2.10).

---

## Author Comment (AC2)

**Response to reviewer comments**

We are sincerely grateful to editor and reviewers for their valuable time spent on reviewing our manuscript. The comments are very helpful and valuable, and we have addressed some issues raised by the reviewers in the revised manuscript. Please find our point-by-point response (in blue font) to the comments (in black font) raised by reviewers.

**Reviewer 2**

This study compared three wind speed distributions of kernel, Weibull, and Rayleigh type for estimating average wind power density under varied desert steppe terrain contexts. Three key parameters of scale factor (c) and shape factor(k) from the Weibull model and surface roughness (z0) were investigated for estimating wind energy resource. Authors pointed that the key parameters (c, k, and z0) should be accurately considered for estimating wind energy resources under varied desert steppe terrain contexts. The work is interesting and informative for wind energy evaluation. The manuscript is well organized but need proofreading by native speakers. I recommend minor revision.

**Response:** Many thanks for your positive comments. We are very grateful for all the constructive comments and suggestions. We have adopted all the suggestions in our revised manuscript.

Line 52. Citations format. "(Chang, 2011a) used six …" should be "Chang (2011a) used six …". The same for other citations.

**Response:** Many thanks for your kind suggestion. Amended.

Line 116. Change "The average daily wind speed" to "The daily averaged wind speed".

**Response:** Many thanks for your kind suggestion. Amended.

Line 200. I think your precision is too high here. A single decimal place is probably all you can state here. The same for line 218.

**Response:** Many thanks for your kind suggestion. Amended.

Table 1 and 2. What is the information of the shading. In fact, it is hard for readers to get information from this kind of table.

**Response:** Thanks for your question. Because table 1 and table 2 are relatively long, the shades of red and blue can be used to represent the size of the value. The red shading and blue shading represent the larger and smaller values in the table, respectively. The darker the color, the more extreme the value. It is convenient to compare with Figure 4 and Figure 6, for example, the size of the scale factor c in

Figure 4 represents the approximate position of the probability density maximum value of the distribution curve.

Figure 2c. A curve of monthly averaged wind speed added is more informative.

**Response:** Thank you very much for your kind suggestion, we have added a curve of monthly averaged wind speed as following.

[Figure]

Figure 4. The x-label should be WS70 (m s-1). Season and year information should texted in each plot.

**Response:** Many thanks for your kind suggestion. Amended.

[Figure]

Figure 5. No need to give x-axis tick-labels in every plot as all plots used one x-axis. The font size need to be unified for all the labels. The word "period" is not needed.

**Response:** Thanks for your suggestions. Amended.

[Figure]

Figure 6. The x-label is not "Altitude", it should be Height (above the ground level). What is the unit for x-label? The x-tick label should be 10, 20, 30, 40, … , 100.

**Response:** Thanks for your suggestion. Amended.